# Maximum Entropy Monte-Carlo Planning

**Chenjun Xiao**[1]    **Jincheng Mei**[1]    **Ruitong Huang**[2]    **Dale Schuurmans**[1]    **Martin Müller**[1]
[1]University of Alberta
[2]Borealis AI
{chenjun, jmei2, daes, mmueller}@ualberta.com, ruitong.huang@borealisai.com

## Abstract

We develop a new algorithm for online planning in large scale sequential decision problems that improves upon the worst case efficiency of UCT. The idea is to augment Monte-Carlo Tree Search (MCTS) with maximum entropy policy optimization, evaluating each search node by softmax values back-propagated from simulation. To establish the effectiveness of this approach, we first investigate the single-step decision problem, stochastic softmax bandits, and show that softmax values can be estimated at an optimal convergence rate in terms of mean squared error. We then extend this approach to general sequential decision making by developing a general MCTS algorithm, *Maximum Entropy for Tree Search* (MENTS). We prove that the probability of MENTS failing to identify the best decision at the root decays exponentially, which fundamentally improves the polynomial convergence rate of UCT. Our experimental results also demonstrate that MENTS is more sample efficient than UCT in both synthetic problems and Atari 2600 games.

## 1 Introduction

Monte Carlo planning algorithms have been widely applied in many challenging problems [12, 13]. One particularly powerful and general algorithm is the Monte Carlo Tree Search (MCTS) [3]. The key idea of MCTS is to construct a search tree of states that are evaluated by averaging over outcomes from simulations. MCTS provides several major advantages over traditional online planning methods. It breaks the curse of dimensionality by simulating state-action trajectories using a domain generative model, and building a search tree online by collecting information gathered during the simulations in an incremental manner. It can be combined with domain knowledge such as function approximations learned either online [17] or offline [12, 13]. It is highly selective, where bandit algorithm are applied to balance between exploring the most uncertain branches and exploiting the most promising ones [9]. MCTS has demonstrated outstanding empirical performance in many game playing problems, but most importantly, it is provable to converge to the optimal policy if the exploitation and exploration balanced appropriately [9, 7].

The convergence property of MCTS highly replies on the state value estimations. At each node of the search tree, the value estimation is also used to calculate the value of the action leading to that node. Hence, the convergence rate of the state value estimation influences the rate of convergence for states further up in the tree. However, the Monte Carlo value estimate (average over simulations outcomes) used in MCTS does not enjoy effective convergence guarantee when this value is back-propagated in the search tree, since for any given node, the sampling policy in the subtree is changing and the payoff sequences experienced will drift in time. In summary, the compounding error, caused by the structure of the search tree as well as the uncertainty of the Monte Carlo estimation, makes that UCT can only guarantee a polynomial convergence rate of finding the best action at the root.

Ideally, one would like to adopt a state value that can be efficiently estimated and back-propagated in a search tree. In this paper, we exploit the usage of *softmax value estimate* in MCTS based on the maximum entropy policy optimization framework. To establish the effectiveness of this approach,

we first propose a new *stochastic softmax bandit* framework for the single-step decision problem, and show that softmax values can be estimated in a sequential manner at an optimal convergence rate in terms of mean squared error. Our next contribution is to extend this approach to general sequential decision making by developing a general MCTS algorithm, *Maximum Entropy for Tree Search* (MENTS). We contribute new observations that the softmax state value can be efficiently back-propagated in the search tree, which enables the search algorithm to achieve faster convergence rate towards finding the optimal action at the root. Our theoretical analysis shows that MENTS enjoys an exponential convergence rate to the optimal solution, improving the polynomial convergence rate of UCT fundamentally. Our experiments also demonstrate that MENTS is much more sample efficient compared with UCT in practice.

## 2    Background

### 2.1    Online Planning in Markov Decision Process

We focus on the episodic Markov decision process (MDP) [1], which is formally defined as a 5-tuple $\{\mathcal{S}, \mathcal{A}, P, R, H\}$. $\mathcal{S}$ is the state space, $\mathcal{A}$ is the action space. $H$ is the maximum number of steps at each episode, $P$ and $R$ are the transition and reward functions, such that $P(\cdot|s, a)$ and $R(s, a)$ give the next state distribution and reward of taking action $a$ at state $s$. We assume the transition and reward functions are deterministic for simplicity, while all of our techniques can easily generalize to the case with stochastic transitions and rewards, with an appropriate dependence on the variances of the transition and reward distributions. The solution of an MDP is a policy $\pi$ that maps any state $s$ to a probability distribution over actions. The optimal policy maximizes, on expectation, the cumulative sum of rewards, defined as,

$$G_t = \sum_{k=0}^{H+1} R_{t+k}, \qquad R_t = \begin{cases} R(s_t, a_t), & t \leq H \\ \nu(s_{H+1}), & t = H + 1 \end{cases}$$

where we assume an oracle function $\nu$ that assigns stochastic evaluations for states at the end of episode. We note that this definition can also be used as a general formulation for planning algorithms in infinite horizon MDP, since $H$ can be considered as the maximum search depth, and a stochastic evaluation function is applied at the end. We assume $\nu$ is subgaussian and has variance $\sigma^2$.

For policy $\pi$, the *state value function* $V^\pi(s)$, is defined to be the expected sum of rewards from $s$, $V^\pi(s) = \mathbb{E}^\pi [G_t | s_t = s]$. The *state-action value function*, also known as the Q-value, is defined similarly, $Q^\pi(s, a) = \mathbb{E}^\pi [G_t | s_t = s, a_t = a]$. The *optimal value functions* are the maximum value achievable by any policy, $V^*(s) = \max_\pi V^\pi(s)$, $Q^*(s, a) = \max_\pi Q^\pi(s, a)$. The *optimal policy* is defined by the greedy policy with respect to $Q^*$, $\pi^*(s) = \text{argmax}_a Q^*(s, a)$. It is well known that the optimal values can be recursively defined by the Bellman optimality equation,

$$Q^*(s, a) = R(s, a) + \mathbb{E}_{s'|s,a} [V^*(s')], \qquad V^*(s) = \max_a Q^*(s, a). \qquad (1)$$

We consider the *online planning* problem that uses a *generative model* of the MDP to compute the optimal policy at any input state, given a fixed sampling budget. The generative model is a randomized algorithm that can output the reward $R(s, a)$ and sample a next state $s'$ from $P(\cdot|s, a)$, given a state-action pair $(s, a)$ as the input. For example, in the game of Go, if the rules of the game are known, the next board state can be predicted exactly after a move. To solve the online planning problem, an algorithm uses the generative model to sample an episode at each round, and proposes an action for the input state after the sampling budget is expended. The performance of an online planning algorithm can be measured by its probability of proposing the optimal action for the state of interest.

### 2.2    Monte Carlo Tree Search and UCT

To solve the online planning task, Monte Carlo Tree Search (MCTS) builds a *look-ahead tree* $\mathcal{T}$ online in an incremental manner, and evaluates states with Monte Carlo simulations [3]. Each node in $\mathcal{T}$ is labeled by a state $s$, and stores a value estimate $Q(s, a)$ and visit count $N(s, a)$ for each action $a$. The estimate $Q(s, a)$ is the mean return of all simulations starting from $s$ and $a$. The root of $\mathcal{T}$ is

labeled by the state of interest. At each iteration of the algorithm, one simulation starts from the root of the search tree, and proceeds in two stages: a *tree policy* is used to select actions while within the tree until a leaf of $\mathcal{T}$ is reached. An evaluation function is used at the leaf to obtain a simulation return. Typical choices of the evaluation function include function approximation with a neural network, and Monte Carlo simulations using a *roll-out policy*. The return is propagated upwards to all nodes along the path to the root. $\mathcal{T}$ is grown by expanding the leaf reached during the simulation.

Bandit algorithms are used to balance between exploring the most uncertain branches and exploiting the most promising ones. The UCT algorithm applies UCB1 as its tree policy to balance the growth of the search tree [9]. At each node of $\mathcal{T}$, its tree policy selects an action with the maximum upper confidence bound

$$\text{UCB}(s,a) = Q(s,a) + c\sqrt{\frac{\log N(s)}{N(s,a)}},$$

where $N(s) = \sum_a N(s,a)$, and $c$ is a parameter controlling exploration. The UCT algorithm has proven to be effective in many practical problems. The most famous example is its usage in AlphaGo [12, 13]. UCT is asymptotically optimal: the value estimated by UCT converges in probability to the optimal value, $Q(s,a) \xrightarrow{p} Q^*(s,a), \forall s \in \mathcal{S}, \forall a \in \mathcal{A}$. The probability of finding a suboptimal action at the root converges to zero at a rate of $O(\frac{1}{t})$, where $t$ is the simulation budget [9].

## 2.3 Maximum Entropy Policy Optimization

The maximum entropy policy optimization problem, which augments the standard expected reward objective with a entropy regularizer, has recently drawn much attention in the reinforcement learning community [4, 5, 11]. Given $K$ actions and the corresponding $K$-dimensional reward vector $\mathbf{r} \in \mathbb{R}^K$, the entropy regularized policy optimization problem finds a policy by solving

$$\max_\pi \left\{ \pi \cdot \mathbf{r} + \tau \mathcal{H}(\pi) \right\}. \tag{2}$$

where $\tau \geq 0$ is a user-specified temperature parameter which controlls the degree of exploration. The most intriguing fact about this problem is that it has a closed form solution. Define the *softmax* $\mathcal{F}_\tau$ and the *soft indmax* $\mathbf{f}_\tau$ functions,

$$\mathbf{f}_\tau(\mathbf{r}) = \exp\{(\mathbf{r} - \mathcal{F}_\tau(\mathbf{r}))/\tau\} \qquad \mathcal{F}_\tau(\mathbf{r}) = \tau \log \sum_a \exp(r(a)/\tau).$$

Note that the softmax $\mathcal{F}_\tau$ outputs a scalar while the soft indmax $\mathbf{f}_\tau$ maps any reward vector $\mathbf{r}$ to a Boltzmann policy. $\mathcal{F}_\tau(\mathbf{r})$, $\mathbf{f}_\tau(\mathbf{r})$ and (2) are connected by as shown in [4, 11],

$$\mathcal{F}_\tau(\mathbf{r}) = \max_\pi \left\{ \pi \cdot \mathbf{r} + \tau \mathcal{H}(\pi) \right\} = \mathbf{f}_\tau(\mathbf{r}) \cdot \mathbf{r} + \tau \mathcal{H}(\mathbf{f}_\tau(\mathbf{r})). \tag{3}$$

This relation suggests the softmax value is an upper bound on the maximum value, and the gap can be upper bounded by the product of $\tau$ and the maximum entropy. Note that as $\tau \to 0$, (2) approaches the standard expected reward objective, where the optimal solution is the hard-max policy. Therefore, it is straightforward to generalize the entropy regularized optimization to define the *softmax value functions*, by replacing the hard-max operator in (1) with the softmax operators [4, 11],

$$Q^*_{\text{sft}}(s,a) = R(s,a) + \mathbb{E}_{s'|s,a}\left[V^*_{\text{sft}}(s')\right], \qquad V^*_{\text{sft}}(s) = \tau \log \sum_a \exp\left\{Q^*_{\text{sft}}(s,a)/\tau\right\}. \tag{4}$$

Finally, according to (3), we can characterize the optimal *softmax policy* by,

$$\pi^*_{\text{sft}}(a|s) = \exp\left\{ \left(Q^*_{\text{sft}}(s,a) - V^*_{\text{sft}}(s)\right)/\tau \right\}. \tag{5}$$

In this paper, we combine the maximum entropy policy optimization framework with MCTS, by estimating the softmax values backpropagated from simulations. Specifically, we show that the softmax values can be efficiently backpropagated in the search tree, which leads to a faster convergence rate to the optimal policy at the root.

## 3 Softmax Value Estimation in Stochastic Bandit

We begin by introducing the *stochastic softmax bandit* problem. We provide an asymptotical lower bound of this problem, propose a new bandit algorithm for it and show a tight upper bound on its convergence rate. Our upper bound matches the lower bound not only in order, but also in the coefficient of the dominating term. All proofs are provided in the supplementary material.

## 3.1 The Stochastic Softmax Bandit

Consider a stochastic bandit setting with arms set $\mathcal{A}$. At each round $t$, a learner chooses an action $A_t \in \mathcal{A}$. Next, the environment samples a random reward $R_t$ and reveals it to the learner. Let $r(a)$ be the expected value of the reward distribution of action $a \in \mathcal{A}$. We assume $r(a) \in [0,1]$, and that all reward distributions are $\sigma^2$-subgaussian [2]. For round $t$, we define $N_t(a)$ as the number of times $a$ is chosen so far, and $\hat{r}_t(a)$ as the empirical estimate of $r(a)$,

$$N_t(a) = \sum\nolimits_{i=1}^{t} \mathbb{I}\{A_t = a\} \qquad \hat{r}_t(a) = \sum\nolimits_{i=1}^{t} \mathbb{I}\{A_i = a\} R_i / N_t(a),$$

where $\mathbb{I}\{\cdot\}$ is the indicator function. Let $\mathbf{r} \in [0,1]^K$ be the vector of expected rewards, and $\hat{\mathbf{r}}_t$ be the empirical estimates of $\mathbf{r}$ at round $t$. We denote $\pi_{\text{sft}}^* = \mathbf{f}_\tau(\mathbf{r})$ the optimal soft indmax policy defined by the mean reward vector $\mathbf{r}$. The stochastic bandit setting can be considered as a special case of an episodic MDP with $H = 1$.

In a stochastic softmax bandit problem, instead of finding the policy with maximum expected reward as in original stochastic bandits [10], our objective is to estimate the softmax value $V_{\text{sft}}^* = \mathcal{F}_\tau(\mathbf{r})$ for some $\tau > 0$. We define $U^* = \sum_a \exp\{r(a)/\tau\}$ and $U_t = \sum_a \exp\{\hat{r}_t(a)/\tau\}$, and propose to use the estimator $V_t = \mathcal{F}_\tau(\hat{\mathbf{r}}_t) = \tau \log U_t$. Our goal is to find a sequential sampling algorithm that can minimize the mean squared error, $\mathcal{E}_t = \mathbb{E}[(U^* - U_t)^2]$. The randomness in $\mathcal{E}_t$ comes from both the sampling algorithm and the observed rewards. Our first result gives a lower bound on $\mathcal{E}_t$.

**Theorem 1.** *In the stochastic softmax bandit problem, for any algorithm that achieves $\mathcal{E}_t = O(\frac{1}{t})$, there exists a problem setting such that*

$$\lim_{t \to \infty} t\mathcal{E}_t \geq \frac{\sigma^2}{\tau^2} \left( \sum_a \exp(r(a)/\tau) \right)^2.$$

*Also, to achieve this lower bound, there must be for any $a \in \mathcal{A}$, $\lim_{t \to \infty} N_t(a)/t = \pi_{\text{sft}}^*(a)$.*

Note that in Theorem 1, we only assume $\mathcal{E}_t = O(1/t)$, but not that the algorithm achieves (asymptotically) unbiased estimates for each arm. Furthermore, this lower bound also reflects the consistency between the softmax value and the soft indmax policy (3): in order to achieve the lower bound on the mean squared error, the sampling policy must converge to $\pi_{\text{sft}}^*$ asymptotically.

## 3.2 E2W: an Optimal Sequential Sampling Strategy

Inspired by the lower bound, we propose an optimal algorithm, Empirical Exponential Weight (E2W), for the stochastic softmax bandit problem. The main idea is very intuitive: enforce enough exploration to guarantee good estimation of $\hat{\mathbf{r}}$, and make the policy converge to $\pi^*$ asymptotically, as suggested by the lower bound. Specifically, at round $t$, the algorithm selects an action by sampling from the distribution

$$\pi_t(a) = (1 - \lambda_t)\mathbf{f}_\tau(\hat{\mathbf{r}})(a) + \lambda_t \frac{1}{|\mathcal{A}|}. \tag{6}$$

In (6), $\lambda_t = \varepsilon|\mathcal{A}|/\log(t+1)$ is a decay rate for exploration, with exploration parameter $\varepsilon > 0$. Our next theorem provides an exact convergence rate for E2W.

**Theorem 2.** *For the softmax stochastic bandit problem, E2W can guarantee,*

$$\lim_{t \to \infty} t\mathcal{E}_t = \frac{\sigma^2}{\tau^2} \left( \sum_a \exp(r(a)/\tau) \right)^2.$$

Theorem 2 shows that E2W is an asymptotically optimal sequential sampling strategy for estimating the softmax value in stochastic multi-armed bandits. The main contribution of the present paper is the introduction of the softmax bandit algorithm for the implementation of tree policy in MCTS. In our proposed new algorithm, softmax bandit is used as the fundamental tool both for estimating each state's softmax value, and balancing the growth of the search tree.

# 4 Maximum Entropy MCTS

We now describe the main technical contributions of this paper, which combine maximum entropy policy optimization with MCTS. Our proposed method, MENTS (Maximum Entropy for Tree Search), applies a similar algorithmic design as UCT (see Section 2.2) with two innovations: using E2W as the tree policy, and evaluating each search node by softmax values back-propagated from simulations.

## 4.1 Algorithmic Design

Let $\mathcal{T}$ be a look-ahead search tree built online by the algorithm. Each node $n(s) \in \mathcal{T}$ is labeled by a state $s$, contains a softmax value estimate $Q_{\text{sft}}(s, a)$, and a visit count $N(s, a)$ for each action $a$. We use $\mathbf{Q}_{\text{sft}}(s)$ to denote a $|\mathcal{A}|$-dimensional vector with components $Q_{\text{sft}}(s, a)$. Let $N(s) = \sum_a N(s, a)$ and $V_{\text{sft}}(s) = \mathcal{F}_\tau(\mathbf{Q}_{\text{sft}}(s))$. During the in-tree phase of the simulation, the tree policy selects an action according to

$$\pi_t(a|s) = (1 - \lambda_s)\mathbf{f}_\tau(\mathbf{Q}_{\text{sft}}(s))(a) + \lambda_s \frac{1}{|\mathcal{A}|} \tag{7}$$

where $\lambda_s = \varepsilon|\mathcal{A}|/\log(\sum_a N(s, a) + 1)$. Let $\{s_0, a_0, s_1, a_1, \ldots, s_T\}$ be the state action trajectory in the simulation, where $n(s_T)$ is a leaf node of $\mathcal{T}$. An evaluation function is called on $s_T$ and returns an estimate $R$ [3]. $\mathcal{T}$ is then grown by expanding $n(s_T)$. Its statistics are initialized by $Q_{\text{sft}}(s_T, a) = 0$ and $N(s_T, a) = 0$ for all actions $a$. For all nodes in the trajectory, we update the visiting counts by $N(s_t, a_t) = N(s_t, a_t) + 1$, and the Q-values using a *softmax backup*,

$$Q_{\text{sft}}(s_t, a_t) = \begin{cases} r(s_t, a_t) + R & t = T - 1 \\ r(s_t, a_t) + \mathcal{F}_\tau(\mathbf{Q}_{\text{sft}}(s_{t+1})) & t < T - 1 \end{cases} \tag{8}$$

The algorithm MENTS can also be extended to use domain knowledge, such as function approximations learned offline. For instance, suppose that a policy network $\tilde{\pi}(\cdot|s)$ is available. Then the statistics can be initialized by $Q_{\text{sft}}(s_T, a) = \log \tilde{\pi}(a|s_T)$ and $N(s_T, a) = 0$ for all actions $a$ during the expansion. Finally, at each time step $t$, MENTS proposes the action with the maximum estimated softmax value at the root $s_0$; i.e. $a_t = \text{argmax}_a Q_{\text{sft}}(s_0, a)$.

## 4.2 Theoretical Analysis

This section provides the key steps in developing a theoretical analysis of the convergence property for MENTS. We first show that for any node in the search tree, after its subtree has been fully explored, the estimated softmax value will converge to the optimal value at an exponential rate. Recall that in Theorem 1, an optimal sampling algorithm for the softmax stochastic bandit problem must guarantee $\lim_{t \to \infty} N_t(a)/t = \pi_{\text{sft}}^*(a)$ for any action $a$. Our first result shows that this holds for true in E2W with high probability. This directly comes from the proof of Theorem 2.

**Theorem 3.** *Consider E2W applied to the stochastic softmax bandit problem. Let $N_t^*(a) = \pi_{sft}^*(a) \cdot t$. Then there exists some constants $C$ and $\tilde{C}$ such that,*

$$\mathbb{P}\left(|N_t(a) - N_t^*(a)| > \frac{Ct}{\log t}\right) \le \tilde{C}|\mathcal{A}|t \exp\left\{-\frac{t}{(\log t)^3}\right\}.$$

In the bandit case, the reward distribution of each arm is assumed to be subgaussian. However, when applying bandit algorithms at the internal nodes of a search tree, the payoff sequence experienced from each action will drift over time, since the sampling probability of the actions in the subtree is changing. The next result shows that even under this drift condition, the softmax value can still be efficiently estimated according to the backup scheme (8).

**Theorem 4.** *For any node $n(s) \in \mathcal{T}$, define the event,*

$$E_s = \left\{\forall a \in \mathcal{A}, |N(s, a) - N^*(s, a)| < \frac{N^*(s, a)}{2}\right\}$$

*where $N^*(s, a) = \pi^*_{sft}(a|s) \cdot N(s)$. For $\epsilon \in [0, 1)$, there exist some constant $C$ and $\tilde{C}$ such that for sufficiently large $t$,*

$$\mathbb{P}\left(|V_{sft}(s) - V^*_{sft}(s)| \geq \epsilon | E_s\right) \leq \tilde{C} \exp\left\{-\frac{N(s)\tau^2\epsilon^2}{C\sigma^2}\right\}.$$

Without loss of generality, we assume $Q^*(s, 1) \geq Q^*(s, 2) \geq \cdots \geq Q^*(s, |\mathcal{A}|)$ for any $n(s) \in \mathcal{T}$, and define $\Delta = Q^*(s, 1) - Q^*(s, 2)$. Recall that by (3), the gap between the softmax and maximum value is upper bounded by $\tau$ times the maximum of entropy. Therefore as long as $\tau$ is chosen small enough such that this gap is smaller than $\Delta$, the best action also has the largest softmax value. Finally, as we are interested in the probability that the algorithm fails to find the optimal arm at the root, we prove the following result.

**Theorem 5.** *Let $a_t$ be the action returned by MENTS at iteration $t$. Then for large enough $t$ with some constant $C$,*

$$\mathbb{P}(a_t \neq a^*) \leq Ct \exp\left\{-\frac{t}{(\log t)^3}\right\}.$$

**Remark.** MENTS enjoys a fundamentally faster convergence rate than UCT. We highlight two main reasons behind this success result from the innovated algorithmic design. First, MENTS applies E2W as the tree policy during simulations. This assures that the softmax value functions used in MENTS could be effectively estimated in an optimal rate, and the tree policy would converge to the optimal softmax policy $\pi^*_{sft}$ asymptotically, as suggested by Theorem 1 and Theorem 2. Secondly, Theorem 4 shows that the softmax value can also be efficiently back-propagated in the search tree. Due to these facts, the probability of MENTS failing to identify the best decision at the root decays exponentially, significantly improving the polynomial rate of UCT.

## 5 Related Work

Maximum entropy policy optimization is a well studied topic in reinforcement learning [4, 5, 11]. The maximum entropy formulation provides a substantial improvement in exploration and robustness, by adopting a smoothed optimization objective and acquiring diverse policy behaviors. The proposed algorithm MENTS is built on the softmax Bellman operator (4), which is used as the value propagation formula in MCTS. To our best knowledge, MENTS is the first algorithm that applies the maximum entropy policy optimization framework for simulation-based planning algorithms.

Several works have been proposed for improving UCT, since it is arguably "over-optimistic" [2] and does not explore sufficiently: UCT can take a long time to discover an optimal branch that initially looked inferior. Previous work has proposed to use flat-UCB, which enforces more exploration, as the tree policy for action selection at internal nodes [2]. Minimizing simple regret in MCTS is discussed in [1, 16]. Instead of using UCB1 as the tree policy at each node, these works employ a hybrid architecture, where a best-arm identification algorithm such as Sequential Halving [6] is applied at the upper levels, while the original UCT is used for the deeper levels of the tree.

Various value back-propagation strategies, particularly back-propagate the maximum estimated value over the children, were originally studied in [3]. It has been shown that the maximum backup is a poor option, since the Monte-Carlo estimation is too noisy when the number of simulations is low, which misguides the algorithm, particularly at the beginning of search. Complex back-propagation strategies in MCTS have been investigated in [8], where a mixture of maximum backup with the well known TD-$\lambda$ operator [15] is proposed. In contrast to these approaches, MENTS exploits the softmax backup to achieve a faster convergence rate of value estimation.

## 6 Experiments

We evaluate the proposed algorithm, MENTS, across several different benchmark problems against strong baseline methods. Our first test domain is a *Synthetic Tree* environment. The tree has branching factor (number of actions) $k$ of depth $d$. At each leaf of the tree, a standard Gaussian distribution is assigned as an evaluation function, that is each time a leaf is visited, the distribution is used to sample a stochastic return. The mean of each Gaussian distribution is determined in the following way: when initializing the environment each edge is assigned a random value, then the mean of the Gaussian

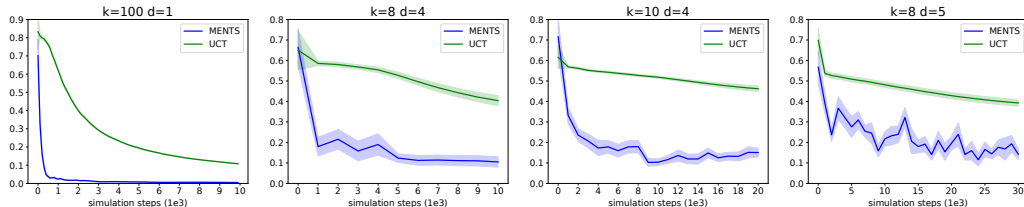

Figure 1: Evaluation of softmax value estimation in the synthetic tree environment. The x-axis shows the number of simulations and y-axis shows the value estimation error. The shaded area shows the standard error. We find that the softmax value can be efficiently estimated by MENTS.

distribution at a leaf is the sum of values along the path from the root to the leaf. This environment is similar to the P-game tree environment [9, 14] used to model two player minimax games, while here we consider the single (max) player version. Finally, we normalize all the means in $[0, 1]$.

We then test MENTS on five Atari games: *BeamRider*, *Breakout*, *Q*bert*, *Seaquest* and *SpaceInvaders*. For each game, we train a vanilla DQN and use it as an evaluation function for the tree search as discussed in the AlphaGo [12, 13]. In particular, the softmax of Q-values is used as the state value estimate, and the Boltzmann distribution over the Q-values is used as the policy network to assign a probability prior for each action when expanding a node. The temperature is set to $0.1$. The UCT algorithm adopts the following tree-policy introduced in AlphaGo [13],

$$\text{PUCT}(s, a) = Q(s, a) + cP(s, a)\frac{\sqrt{\sum_b N(s, b)}}{1 + N(s, a)}$$

where $P(s, a)$ is the prior probability. MENTS also applies the same evaluation function. The prior probability is used to initialize the $Q_{\text{sft}}$ as discussed in Section 4.1. We note that the DQN is trained using a hard-max target. Training a neural network using softmax targets such as soft Q-learning or PCL might be more suitable for MENTS [4, 11]. However, in the experiments we still use DQN in MENTS to present a fair comparison with UCT, since both algorithms apply the exactly same evaluation function. The details of the experimental setup are provided in the Appendix.

## 6.1 Results

**Value estimation in synthetic tree**. As shown in Section 4.2, the main advantage of the softmax value is that it can be efficiently estimated and back-propagated in the search tree. To verify this observation, we compare the value estimation error of MENTS and UCT in both the bandit and tree search setting. For MENTS, the error is measured by the absolute difference between the estimated softmax value $V_{\text{sft}}(s_0)$ and the true softmax state value $V_{\text{sft}}^*(s_0)$ of the root $s_0$. For UCT, the error is measured by the absolute difference between the Monte Carlo value estimation $V(s_0)$ and the optimal state value $V^*(s_0)$ at the root. We report the results in Figure 1. Each data point is averaged over $5 \times 5$ independent experiment (5 runs on 5 randomly initialized environment). In all of the test environments, we observe that MENTS estimates the softmax values efficiently. By comparison, we find that the Monte Carlo estimation used in UCT converges far more slowly to the optimal state value, even in the bandit setting $(d = 1)$.

**Online planning in synthetic tree**. We next compare MENTS with UCT for online planning in the synthetic tree environment. Both algorithms use Monte Carlo simulation with uniform rollout policy as the evaluation function. The error is evaluated by $V^*(s_0) - Q^*(s_0, a_t)$, where $a_t$ is the action proposed by the algorithm at simulation step $t$, and $s_0$ is the root of the synthetic tree. The optimal values $Q^*$ and $V^*$ are computed by back-propagating the true values from the leaves when the environment is initialized. Results are reported in Figure 2. As in the previous experiment, each data point is averaged over $5 \times 5$ independent experiment (5 runs on 5 randomly initialized environment). UCT converges faster than our method in the bandit environment $(d = 1)$. This is because that the main advantage of MENTS is the usage of softmax state values, which can be efficiently estimated and back-propagated in the search tree. In the bandit case such an advantage does not exist. In the tree case $(d > 0)$, we find that MENTS significantly outperforms UCT, especially in the large domain. For example, in synthetic tree with $k = 8$ $d = 5$, UCT fails to identify the optimal action at the root in some of the random environments, result in the large regret given the simulation budgets. However,

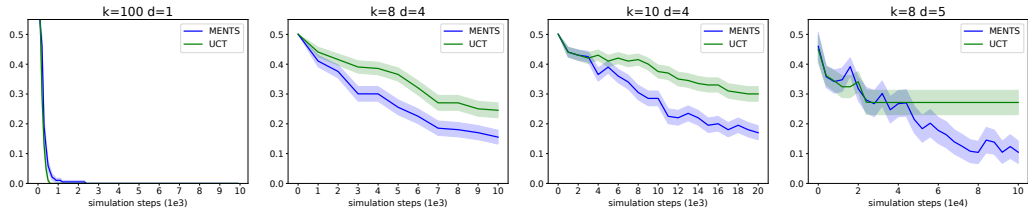

Figure 2: Evaluation of online planning in the synthetic tree environment. The x-axis shows the number of simulations and y-axis shows the planning error. The shaded area shows the standard error. We can observe that MENTS enjoys much smaller error than UCT especially in the large domain.

Table 1: Performance comparison of Atari games playing.

| Agent | *BeamRider* | *Breakout* | *Q\*bert* | *Seaquest* | *SpaceInvaders* |
|---|---|---|---|---|---|
| **DQN** | 19280 | 345 | 14558 | 1142 | 625 |
| **UCT** | 21952 | 367 | 16010 | 1129 | 656 |
| **MENTS** | 18576 | **386** | **18336** | 1161 | **1503** |

MENTS can continuously make progress towards the optimal solution in all random environments, confirming MENTS scales with larger tree depth.

**Online planning in Atari 2600 games**. Finally, we compare MENTS and UCT using Atari games. At each time step we use 500 simulations to generate a move. Results are provided in Table 1, where we highlight scores where MENTS significantly outperforms the baselines. Scores obtained by DQN are also provided. In *Breakout*, *Q\*bert* and *SpaceInvaders*, MENTS significantly outperforms UCT as well as the DQN agent. In *BeamRider* and *Seaquest* all algorithms performs similarly, since the search algorithms only use the DQN as the evaluation function and only 500 simulations are applied to generate a move. We can expect better performance when a larger simulation budget is used.

# 7 Conclusion

We propose a new online planning algorithm, Maximum Entropy for Tree Search (MENTS), for large scale sequential decision making. The main idea of MENTS is to augment MCTS with maximum entropy policy optimization, evaluating each node in the search tree using softmax values back-proagated from simulations. We contribute two new observations that are essential to establishing the effectiveness of MENTS: first, we study *stochastic softmax bandits* for single-step decision making and show that softmax values can be estimated at an optimal convergence rate in terms of mean squared error; second, the softmax values can be efficiently back-propagated from simulations in the search. We prove that the probability of MENTS failing to identify the best decision at the root decays exponentially, which fundamentally improves the worst case efficiency of UCT. Empirically, MENTS exhibits a significant improvement over UCT in both synthetic tree environments and Atari game playing.

# Acknowledgement

The authors wish to thank Csaba Szepesvari for useful discussions, and the anonymous reviewers for their valuable advice. Part of the work is performed when the first two authors were interns at BorealisAI. This research was supported by NSERC, the Natural Sciences and Engineering Research Council of Canada, and AMII, the Alberta Machine Intelligence Institute.

## Footnotes

[1]All of our approaches can extend to infinite horizon MDP.

[2]For prudent readers, we follow the finite horizon bandits setting in [10], where the probability space carries the tuple of random variables $S_T = \{A_0, R_0, \ldots, A_T, R_T\}$. For every time step $t-1$ the historical observation defines a $\sigma$-algebra $\mathcal{F}_{t-1}$ and $A_t$ is $\mathcal{F}_{t-1}$-measurable, the conditional distribution of $A_t$ is our policy at time $\pi_t$, and the conditoinal distribution of the reward $R_{A_t} - r(A_t)$ is a martingale difference sequence.

[3]We adapt a similar setting to Section 3, where $R_t$ is replaced by the sample from the evaluation function, and the martingale assumption is extended to the the selection policy and the evaluation function on the leaves.

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
