[Supplementary Material · neurips2019_ments_full.pdf]

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

## A  Experimental Details

We provide the experiment details in this section.

**Value estimation in synthetic tree**. For all settings, we use $\tau = 0.01$ for the softmax value. The exploration parameters for both MENTS and UCT are tuned from $\{0.1, 0.2, 0.4, 0.6, 0.8, 1.0, 1.5, 2.0\}$.

**Online planning in synthetic tree**. The exploration parameters for MENTS and UCT are tuned from $\{0.1, 0.2, 0.4, 0.6, 0.8, 1.0, 1.5, 2.0\}$. The temperature parameter $\tau$ of MENTS is tuned from $\{0.5, 0.1, 0.05, 0.01, 0.005\}$.

**Online planning in Atari 2600 games**. The exploration parameter for both algorithms are tuned from $\{5.0, 2.0, 1.0, 0.5, 0.1\}$ The temperature parameter $\tau$ of MENTS is tuned from $\{0.1, 0.05, 0.01\}$. The results is averaged over ten environment restarts.

In games such as *BeamRider*, one test game will take thousands of environment steps. Therefore, we only test the algorithms within 10,000 environment steps. The search algorithms are used every 10 steps. For the other steps the agent will use the DQN to select action.

## B  Proofs for softmax stochastic bandit

We first introduce a Lemma that approximates the exponential function of empirical estimator using delta method [1]. This Lemma will be used for both lower bound and upper bound analysis.

**Lemma 1.** *Let $X_1, \ldots, X_n$ be i.i.d. random variables, such that $\mathbb{E}[X_i] = \mu$ and $\mathbb{V}[X_i] = \sigma^2 < \infty$, $\bar{X}_n = \sum_{i=1}^{n} X_i / n$. The first two moment of $\exp\left(\bar{X}_n / \tau\right)$ could be approximated by,*

$$\mathbb{E}\left[\exp\left(\frac{\bar{X}_n}{\tau}\right)\right] = e^{\mu/\tau} + \frac{\sigma^2}{2n}\left(\frac{e^{\mu/\tau}}{\tau^2}\right) + R(n) \tag{9}$$

$$\mathbb{V}\left[\exp\left(\frac{\bar{X}_n}{\tau}\right)\right] = \frac{\sigma^2}{n}\left(\frac{e^{\mu/\tau}}{\tau}\right)^2 + R'(n) \tag{10}$$

*where $|R(n)| \leq O\left(n^{-2}\right), |R'(n)| \leq O\left(n^{-2}\right)$.*

*Proof.* By Taylor's expansion,

$$\exp\left(\frac{\bar{X}_n}{\tau}\right) = e^{\mu/\tau} + \frac{e^{\mu/\tau}}{\tau}\left(\bar{X}_n - \mu\right) + \frac{e^{\mu/\tau}}{2\tau^2}\left(\bar{X}_n - \mu\right)^2 + \frac{e^{\xi/\tau}}{6\tau^3}\left(\bar{X}_n - \mu\right)^3$$

for some $\xi$ between $\mu$ and $\bar{X}_n$. Taking the expectation on both sides,

$$\mathbb{E}\left[\exp\left(\frac{\bar{X}_n}{\tau}\right)\right] = e^{\mu/\tau} + 0 + \frac{e^{\mu/\tau}}{2\tau^2}\mathbb{V}\left[\bar{X}_n\right] + \frac{e^{\xi/\tau}}{6\tau^3}\mathbb{E}\left[\left(\bar{X}_n - \mu\right)^3\right].$$

Let $R(n) = \frac{e^{\xi/\tau}}{6\tau^3}\mathbb{E}\left[\left(\bar{X}_n - \mu\right)^3\right]$. By Lemma 5.3.1 of [1], $|R(n)| \leq O(n^{-2})$, which gives Eq. (9).

Furthermore, note that

$$\left(\mathbb{E}\left[\exp\left(\frac{\bar{X}_n}{\tau}\right)\right]\right)^2 = \left(e^{\mu/\tau} + \frac{\sigma^2}{2n}\left(\frac{e^{\mu/\tau}}{\tau^2}\right) + R(n)\right)^2$$

$$= e^{2\mu/\tau} + \frac{\sigma^2}{n}\left(\frac{e^{\mu/\tau}}{\tau}\right)^2 + \frac{C_1}{n^2}$$

$$+ C_2 R(n) + C_3 \frac{R(n)}{n}$$

for some constant $C_1, C_2, C_3$. On the other hand, following the same idea of deriving Eq. (9),

$$\mathbb{E}\left[\left(\exp\left(\frac{\bar{X}_n}{\tau}\right)\right)^2\right] = e^{2\mu/\tau} + \frac{2\sigma^2}{n}\left(\frac{e^{\mu/\tau}}{\tau}\right)^2 + \tilde{R}(n)$$

where $|\tilde{R}(n)| \leq O(n^{-2})$. The proof of Eq. (10) ends by taking the difference of the above two equations. $\qquad\square$

## B.1  Proof of Theorem 1

We consider the learning problem in a Bayesian setting. In the stochastic bandit problem, we assume the expected reward of each action $r(a)$ is independently sampled from a Gaussian prior $\mathcal{N}(0, \sigma_0^2)$. At time step $t$, for any action $a$, a reward $X_{a,t}$ is sampled from $\mathcal{N}(r(A_t), \sigma^2)$, independently to all the previous observations. The learner chooses an action $A_t$ according to some policy and observe $X_t = X_{A_t,t}$. Without loss of generality, we assume that $\sigma^2 = 1$ and $\tau = 1$. Our goal is to prove

$$\lim_{t\to\infty} \mathbb{E}\left[t\left(U - \hat{U}_t\right)^2 - \frac{\sigma^2}{\tau^2}\left(\sum_a e^{r(a)/\tau}\right)^2\right] \geq 0,$$

where the expectation is taken on the randomness of the algorithm, the expected rewards $\mathbf{r}$, and the observation $X_{a,i}$ given $\mathbf{r}$. Therefore the existence of $\mathbf{r}$ that provides the lower bound is guaranteed since $\mathbf{r}$ satisfies the property in expectation.

We define $\tilde{U}_t$ to be the posterior mean of $U$, i.e. the conditional expectation of $U$ given the observations $X_{a,t}$. Thus, $\mathbb{E}\left[\left(U - \hat{U}_t\right)^2 - \left(U - \tilde{U}_t\right)^2\right] \geq 0$. The benefit of considering $\tilde{U}_t$ is that $\tilde{U}_t$ can further be decomposed into the Bayes estimator of each action, even without the assumption that $\hat{U}_t$ is decomposable or $\hat{U}_t$ has (asymptotic) unbiased estimator for each arm.

We next introduce two technical lemmas that are useful to prove the lower bound. The first result shows that for an algorithm that performs well on all possible environments, it must pull each arm at least in $\Omega(\log t)$ in $t$ rounds. Note that unlike in the regret analysis for stochastic multi-armed bandits, where one only cares about how many times the suboptimal arms are pulled, the $\Omega(\log t)$ lower bound on $N_t(a)$ for suboptimal arms is not strong enough to provides a tight lower bound of $\mathcal{E}_t$.

**Lemma 2.** *For any algorithm $\mathcal{A}$ such that $\mathcal{E}_t = O(\frac{1}{t})$, it holds that $N_t(a) = \Omega(\log t)$ for any arm $a$.*

In the Bayesian learning setting defined above, since $\exp(X_{a,t})$ has a log-normal distribution with a Gaussian prior, its posterior estimation is still log-normal. The second result studies the concentration rate of the posterior estimation.

**Lemma 3.** *Let $\Phi(a) = \frac{\sum_{i=1}^{N_t(a)} X_{a,i} + 1/2}{\tau_0 + N_t(a)}$ be the posterior estimation of $r(a)$ and define $\Delta(a) = e^{r(a)} - e^{\Phi(a)}$. We have*

$$\mathbb{E}\left[\Delta(a)|N_t(a), \mathbf{r}\right] = O\left(\frac{1}{N_t(a)}\right)$$

$$\mathbb{E}\left[\Delta(a)^2 \mid N_t(a), \mathbf{r}\right] = e^{2r(a)}\left(\frac{N_t(a)}{(N_t(a) + \sigma_0)^2} + O\left(\frac{1}{N_t^2(a)}\right)\right).$$

Now we are ready to present the proof of the lower bound.

*Proof of Theorem 1.* By the tower rule and the fact that $\tilde{U}$ is the minimizer of the mean squared error,

$$\mathbb{E}\left[t\left(U - \hat{U}_t\right)^2\right] \geq \mathbb{E}\left[t\left(U - \tilde{U}_t\right)^2\right] = \mathbb{E}\left[\mathbb{E}\left[t\left(U - \tilde{U}_t\right)^2 \,\Big|\, \mathbf{r}\right]\right],$$

It then suffices to prove that

$$\lim_{t\to\infty} \mathbb{E}\left[t\left(U - \tilde{U}_t\right)^2 \,\Big|\, \mathbf{r}\right] \geq \left(\sum_a e^{r(a)}\right)^2$$

for any $\mathbf{r}$. The rest of the proof is always conditioned on $\mathbf{r}$. Let $\mathbf{X}_{a,t} = X_{a,1}, \ldots, X_{a,N_t(a)}$ be the observations of action $a$ up to time step $t$. We can decompose $\tilde{U}$ by

$$\tilde{U}_t = \mathbb{E}\left[U \mid \mathbf{X}_{j,t}, j \in \{1, \ldots, K\}\right] = \sum_{j=1}^{K} \mathbb{E}\left[e^{r(j)} \,\Big|\, \mathbf{X}_{j,t}, j \in \{1, \ldots, K\}\right] = \sum_{j=1}^{K} \mathbb{E}\left[e^{r(j)} \,\Big|\, \mathbf{X}_{j,t}\right].$$

Therefore, the Bayesian estimator of $U$ is

$$\tilde{U}_t = \sum_j \exp\left(\frac{\sum_{i=1}^{N_t(j)} X_{j,i} + 1/2}{\tau_0 + N_t(j)}\right).$$

It remains to bound $\left(U - \tilde{U}_t\right)^2$ conditioned on $\mathbf{r}$. Note that

$$\left(U - \tilde{U}_t\right)^2 = \left(\sum_j e^{r(j)} - \exp\left(\frac{\sum_{k=1}^{N_t(j)} X_{j,k} + 1/2}{\tau_0 + N_t(j)}\right)\right)^2 = \sum_j \Delta_j^2 + \sum_{i \neq j} \Delta_j \Delta_i,$$

where $\Delta_j = e^{r(j)} - \exp(\frac{\sum_{k=1}^{N_t(j)} X_{j,k} + 1/2}{\tau_0 + N_t(j)})$. Finally, define $P_t(j) = N_t(j)/t$ and let $\tau_0 \to 0$. By Lemma 3, we have

$$\lim_{t\to\infty} t\mathbb{E}\left[\left(U - \tilde{U}_t\right)^2 \mid \mathbf{r}\right] = \lim_{t\to\infty} t\mathbb{E}\left[\mathbb{E}\left[\left(U - \tilde{U}_t\right)^2 \mid N_t(1), \dots, N_t(k), \mathbf{r}\right]\right]$$

$$= \lim_{t\to\infty} \mathbb{E}\left[\sum_j \frac{e^{2r(j)} + O\left(\frac{1}{N_t(j)}\right)}{P_t(j)}\right]$$

$$\geq \left(\sum_a e^{r(a)}\right)^2$$

where the last inequality follows by Cauchy-Schwarz inequality and Lemma 2. Note that for the inequality to hold there must be for all action $k \in [K]$, $N_t(k) = N_t^*(k)$.

For the general case, where $\sigma, \tau \neq 1$, we can simply scale the reward by $\tau$, then the variance of $X_{j,k}$ is $\frac{\sigma^2}{\tau^2}$. The proof still holds and we obtain the following inequality,

$$\lim_{t\to\infty} t\mathbb{E}\left[\left(U - \tilde{U}_t\right)^2 \mid \mathbf{r}\right] \geq \frac{\sigma^2}{\tau^2}\left(\sum_a \bar{\pi}(a) e^{r(a)/\tau}\right)^2.$$

$\square$

## B.2 Concentration of $N_t(a)$ in Bandit (Theorem 3)

Define $\tilde{N}_t(a) = \sum_s \pi_s(a)$, where $\pi_s$ is the policy followed by E2W at time step $s$. By Theorem 2.3 in [18] or [19], we have the following concentration result.

$$\mathbb{P}\left(|N_t(a) - \tilde{N}_t(a)| > \epsilon\right) \leq 2\exp\left(-\frac{\epsilon^2}{2\sum_{s=1}^t \sigma_s^2}\right) \leq 2\exp\left(-\frac{2\epsilon^2}{t}\right),$$

where $\sigma_s^2 \leq 1/4$ is the variance of Benoulli distribution with $p = \pi_s(k)$ at time step $s$. Denote the event

$$\widetilde{E}_\epsilon = \{\forall a \in \mathcal{A}, |\tilde{N}_t(a) - N_t(a)| < \epsilon\}.$$

Thus we have

$$\mathbb{P}\left(\widetilde{E}_\epsilon^c\right) \leq 2|\mathcal{A}|\exp\left(-\frac{2\epsilon^2}{t}\right).$$

It remains to bound $\mathbb{P}\left(|\tilde{N}_t(a) - N_t^*(a)| \geq \epsilon\right)$. To prove Theorem 3, we first introduce two technical lemmas, which prove the accuracy of our estimate on the reward and connect the convergence of the reward estimation to the convergence of policy.

**Lemma 4.** *For the stochastic softmax bandit problem, E2W can guarantee that, for $t \geq 4$,*

$$\mathbb{P}\left(\|\mathbf{r} - \hat{\mathbf{r}}_t\|_\infty \geq \frac{2\sigma}{\log(2+t)}\right) \leq 4|\mathcal{A}| \exp\left(-\frac{t}{(\log(2+t))^3}\right).$$

**Lemma 5.** *Given two soft indmax policies, $\pi^{(1)} = \mathbf{f}_\tau(\mathbf{r}^{(1)})$ and $\pi^{(2)} = \mathbf{f}_\tau(\mathbf{r}^{(2)})$, we have*

$$\left\|\pi^{(1)} - \pi^{(2)}\right\|_\infty \leq \left(1 + \frac{1}{\tau}\right)\left\|\mathbf{r}^{(1)} - \mathbf{r}^{(2)}\right\|_\infty$$

**Proof of Theorem 3**

*Proof.* We denote the following event,

$$E_{\mathbf{r}_t} = \left\{\|\mathbf{r} - \hat{\mathbf{r}}_t\|_\infty < \frac{2\sigma}{\log(2+t)}\right\}.$$

For any time step $s$ and action $a$, by the definition of $\pi_s(a)$ we have,

$$|\pi_s(a) - \pi^*(a)| \leq |\hat{\pi}_s(a) - \pi^*(a)| + \lambda_s.$$

Thus, to bound $|\tilde{N}_t(a) - N_t^*(a)|$, conditioned on the event $\cap_{i=1}^t E_{\mathbf{r}_t}$ and for $t \geq 4$ there is,

$$|\tilde{N}_t(a) - N_t^*(a)| \leq \sum_{s=1}^t |\hat{\pi}_s(a) - \pi^*(a)| + \sum_{s=1}^t \lambda_s$$

$$\leq \left(1 + \frac{1}{\tau}\right)\sum_{s=1}^t \|\hat{\mathbf{r}}_s - \mathbf{r}\|_\infty + \sum_{s=1}^t \lambda_s \qquad \text{(by Lemma 5)}$$

$$\leq \left(1 + \frac{1}{\tau}\right)\sum_{s=1}^t \frac{2\sigma}{\log(2+s)} + \sum_{s=1}^t \lambda_s \qquad \text{(by Lemma 4)}$$

$$\leq \left(1 + \frac{1}{\tau}\right)\int_{s=0}^t \frac{2\sigma}{\log(2+s)}\mathrm{d}s + \int_{s=0}^t \frac{|\mathcal{A}|}{\log(1+s)}\mathrm{d}s$$

$$\leq \frac{Ct}{\log t},$$

for some constant $C$ depending on $|\mathcal{A}|$, $\sigma$ and $\tau$. Finally,

$$\mathbb{P}\left(|\tilde{N}_t(a) - N_t^*(a)| \geq \frac{Ct}{\log t}\right) \leq \sum_{i=1}^t \mathbb{P}\left(E_{\mathbf{r}_t}^c\right) = \sum_{i=1}^t 4|\mathcal{A}|\exp\left(-\frac{t}{(\log(2+t))^3}\right)$$

$$\leq 4|\mathcal{A}|t\exp\left(-\frac{t}{(\log(2+t))^3}\right).$$

Therefore,

$$\mathbb{P}\left(|N_t(a) - N_t^*(a)| \geq (1+C)\frac{t}{\log t}\right)$$

$$\leq \mathbb{P}\left(|\tilde{N}_t(k) - N_t^*(k)| \geq \frac{Ct}{\log t}\right) + \mathbb{P}\left(|N_t(k) - \tilde{N}_t(k)| > \frac{t}{\log t}\right)$$

$$\leq 4|\mathcal{A}|t\exp\left(-\frac{t}{\log(2+t)^3}\right) + 2|\mathcal{A}|\exp\left(-\frac{2t}{\log(2+t)^2}\right)$$

$$\leq O\left(t\exp\left(-\frac{t}{(\log t)^3}\right)\right)$$

$\square$

## B.3  Proof of Theorem 2

*Proof of Theorem 2.* Let $\delta_t = Ct/\log t$ with some constant $C$. Define the following set

$$\mathcal{G}_t = \left\{ s \middle| s \in 1 : t, \lceil N_t^*(a) + \delta_t \rceil \geq s \geq \lfloor N_t^*(a) - \delta_t \rfloor \right\},$$

and its complementary set $\mathcal{G}_t^c = \{1, 2, \ldots, t\} \setminus \mathcal{G}_t$.

By Theorem 3, $\forall a \in \{1, \ldots, K\}$, with probability at least $1 - O\left(t \exp\left(-\frac{t}{(\log t)^3}\right)\right)$, $N_t(a) \in \mathcal{G}_t$.
By law of total expectation and Lemma 1,

$$
\begin{aligned}
\mathbb{E}\left[\exp\left(\frac{\hat{r}_t(a)}{\tau}\right)\right] &= \sum_{s=1}^{t} \mathbb{P}\left(N_t(a) = s\right) \mathbb{E}\left[\exp\left(\frac{\hat{r}_t(a)}{\tau}\right) \middle| N_t(a) = s\right] \\
&= \sum_{s=1}^{t} \mathbb{P}\left(N_t(a) = s\right) \left(e^{r(a)/\tau} + \frac{\sigma^2}{2s}\left(\frac{e^{r(a)/\tau}}{\tau^2}\right)\right) + \sum_{s=1}^{t} \mathbb{P}\left(N_t(a) = s\right) O\left(s^{-2}\right) \\
&= \sum_{s=1}^{t} \mathbb{P}\left(N_t(a) = s\right) \left(\frac{\sigma^2}{2s}\left(\frac{e^{r(a)/\tau}}{\tau^2}\right) + O\left(s^{-2}\right)\right) + e^{r(a)/\tau}
\end{aligned}
$$

$$(11)$$

We divide the summation in two parts. For $s \in \mathcal{G}_t^c$, by Theorem 3,

$$\sum_{s \in \mathcal{G}_t^c} \mathbb{P}\left(N_t(a) = s\right) \cdot \left(\frac{\sigma^2}{2s}\left(\frac{e^{r(a)/\tau}}{\tau^2}\right) + O\left(s^{-2}\right)\right) \leq O\left(\frac{1}{t}\right) \tag{12}$$

For $s \in \mathcal{G}_t$,

$$\sum_{s \in \mathcal{G}_t} \mathbb{P}\left(N_t(a) = s\right) \cdot \left(\frac{\sigma^2}{2s}\left(\frac{e^{r(a)/\tau}}{\tau^2}\right) + O\left(s^{-2}\right)\right) \leq O\left((N_t^*(a) - \delta_t)^{-1}\right) \tag{13}$$

Combine the above together,

$$
\begin{aligned}
t\left(U - \mathbb{E}[U_t]\right)^2 &= t\left(\sum_a \mathbb{E}\left[\exp\left(\frac{\hat{r}_t(a)}{\tau}\right)\right] - \exp\left(\frac{r_t(a)}{\tau}\right)\right)^2 \\
&= t\left(\sum_a O\left(\frac{1}{t}\right) + O\left((N_t^*(a) - \delta_t)^{-1}\right)\right)^2.
\end{aligned}
$$

Thus,

$$\lim_{t \to \infty} t\left(U^* - \mathbb{E}[U_t]\right)^2 = 0,$$

i.e. $U_t$ is a consistent estimate for $U^*$.

To bound $\mathcal{E}_t$, it remains to bound the variance of $U_t$ since it is unbiased. By the law of total variance,

$$\mathbb{V}\left[\exp\left(\frac{\hat{r}_t(a)}{\tau}\right)\right] = \mathbb{E}\left[\mathbb{V}\left[\exp\left(\frac{\hat{r}_t(a)}{\tau}\right) \middle| N_t(a)\right]\right] + \mathbb{V}\left[\mathbb{E}\left[\exp\left(\frac{\hat{r}_t(a)}{\tau}\right) \middle| N_t(a)\right]\right] \tag{14}$$

Note that by Lemma 1, the first term is

$$
\begin{aligned}
&\mathbb{E}\left[\mathbb{V}\left[\exp\left(\frac{\hat{r}_t(a)}{\tau}\right) \middle| N_t(a)\right]\right] \\
&= \sum_{s=1}^{t} \mathbb{P}\left(N_t(a) = s\right) \mathbb{V}\left[\exp\left(\frac{\hat{r}_t(a)}{\tau}\right) \middle| N_t(a) = s\right] \\
&= \sum_{s=1}^{t} \mathbb{P}\left(N_t(a) = s\right) \left(\frac{\sigma^2}{s}\left(\frac{e^{r(a)/\tau}}{\tau}\right)^2 + O\left(s^{-\frac{3}{2}}\right)\right)
\end{aligned}
$$

Using the same idea in Eq. (12) and Eq. (13), we consider the summation in two parts. For $s \in \mathcal{G}_t^c$,

$$\sum\nolimits_{s \in \mathcal{G}_t^c} \mathbb{P}\left(N_t(a) = s\right) \cdot \left(\frac{\sigma^2}{s}\left(\frac{e^{r(a)/\tau}}{\tau}\right)^2 + O\left(s^{-\frac{3}{2}}\right)\right) \leq O\left(\frac{1}{t}\right)$$

For $s \in \mathcal{G}_t$,

$$\sum\nolimits_{s \in \mathcal{G}_t} \mathbb{P}\left(N_t(a) = s\right) \cdot \left(\frac{\sigma^2}{s}\left(\frac{e^{r(a)/\tau}}{\tau}\right)^2 + O\left(s^{-\frac{3}{2}}\right)\right) \leq \frac{\sigma^2}{\tau^2} \cdot \frac{e^{2r(a)/\tau}}{N_t^*(a) - \delta_t} + O\left((N_t^*(a) - \delta_t)^{-\frac{3}{2}}\right)$$

Put these together we have,

$$\mathbb{E}\left[\mathbb{V}\left[\exp\left(\frac{\hat{r}_t(a)}{\tau}\right)\Big| N_t(a)\right]\right] \leq O\left(\frac{1}{t}\right) + \frac{\sigma^2}{\tau^2} \cdot \frac{e^{2r(a)/\tau}}{N_t^*(a) - \delta_t} + O\left((N_t^*(a) - \delta_t)^{-\frac{3}{2}}\right) \quad (15)$$

For the second term of Eq. (14) we have,

$$\mathbb{V}\left[\mathbb{E}\left[\exp\left(\frac{\hat{r}_t(a)}{\tau}\right)\Big| N_t(a)\right]\right] = \mathbb{E}\left[\left(\mathbb{E}\left[\exp\left(\frac{\hat{r}_t(a)}{\tau}\right)\Big| N_t(a)\right]\right)^2\right] - \left(\mathbb{E}\left[\exp\left(\frac{\hat{r}_t(a)}{\tau}\right)\right]\right)^2$$

For the first term, by Lemma 1,

$$\mathbb{E}\left[\left(\mathbb{E}\left[\exp\left(\frac{\hat{r}_t(a)}{\tau}\right)\Big| N_t(a)\right]\right)^2\right]$$

$$= \sum_{s=1}^{t} \mathbb{P}\left(N_t(a) = s\right)\left(\mathbb{E}\left[\exp\left(\frac{\hat{r}_t(a)}{\tau}\right)\Big| N_t(a)\right]\right)^2$$

$$= \sum_{s=1}^{t} \mathbb{P}\left(N_t(a) = s\right)\left(e^{2r(a)/\tau} + \frac{\sigma^2}{s}\left(\frac{e^{r(a)/\tau}}{\tau}\right)^2\right) + O\left(s^{-3/2}\right)$$

$$\leq e^{2r(a)/\tau} + O\left(\frac{1}{t}\right) + \frac{\sigma^2}{\tau^2} \cdot \frac{e^{2r(a)/\tau}}{N_t^*(a) - \delta_t} + O\left((N_t^*(a) - \delta_t)^{-\frac{3}{2}}\right)$$

where the last inequality follows by the same idea of proving (15). For the second term, combining Eqs. (11) to (13),

$$\left(\mathbb{E}\left[\exp\left(\frac{\hat{r}_t(a)}{\tau}\right)\right]\right)^2 = \exp\left(\frac{2r(a)}{\tau}\right) + O\left(\frac{1}{t}\right) + O\left((N_t^*(a) - \delta_t)^{-1}\right)$$

Then we have,

$$\mathbb{V}\left[\mathbb{E}\left[\exp\left(\frac{\hat{r}_t(a)}{\tau}\right)\Big| N_t(a)\right]\right] \leq O\left(\frac{1}{t}\right) + \frac{\sigma^2}{\tau^2} \cdot \frac{e^{2r(a)/\tau}}{N_t^*(a) - \delta_t} + O\left((N_t^*(a) - \delta_t)^{-1}\right) \quad (16)$$

Note that

$$\lim_{t\to\infty} t \cdot \frac{\sigma^2}{\tau^2} \cdot \frac{e^{2r(a)/\tau}}{N_t^*(a) - \delta_t} = \lim_{t\to\infty} \frac{\sigma^2}{\tau^2} \cdot \frac{e^{2r(a)/\tau}}{\pi^*(a) - \frac{\delta_t}{t}}$$

$$= \frac{\sigma^2}{\tau^2} \cdot \frac{e^{r(a)/\tau}}{\bar{\pi}(a)} \cdot \left(\sum_a \bar{\pi}(a)\exp(r(a)/\tau)\right) \quad (17)$$

Combine Eq. (15), Eq. (16) and Eq. (17) together,

$$
\lim_{t\to\infty} t\mathbb{V}\left[\hat{U}_t\right]
$$

$$
= \lim_{t\to\infty} t\left(\sum_a \bar{\pi}^2(a)\mathbb{V}\left[\exp\left(\frac{\hat{r}_t(a)}{\tau}\right)\right]\right)
$$

$$
\leq \lim_{t\to\infty} t\sum_a \bar{\pi}^2(a)\left(O\left(\frac{1}{t}\right) + \frac{\sigma^2}{\tau^2}\cdot\frac{e^{2r(a)/\tau}}{N_t^*(a) - \delta_t}\right)
$$

$$
+ t\sum_a \bar{\pi}^2(a)O\left((N_t^*(a) - \delta_t)^{-1}\right)
$$

$$
= \frac{\sigma^2}{\tau^2}\left(\sum_a \bar{\pi}(a)e^{r(a)/\tau}\right)^2
$$

which ends the proof. $\qquad\square$

## B.4 Technical Lemmas

*Proof of Lemma 2.* Consider two gaussian environments $\nu_1$ and $\nu_2$ with unit variance. The vector of means of the reward per arm in $\nu_1$ is $(r(1),\ldots,r(K))$ and $(r(1)+2\epsilon, r(2),\ldots,r(K))$ in $\nu_2$. Define

$$
U_1 = \sum_{i=1}^K e^{r_i}, \quad U_2 = e^{r_1+2\epsilon} + \sum_{i=2}^K e^{r_i}
$$

Let $\mathbb{P}_1$ and $\mathbb{P}_2$ be the distribution induced by $\nu_1$ and $\nu_2$ respectively. Denote the event,

$$
E = \left\{|\hat{U}_t - U_1| > e^{r_1}\epsilon\right\},
$$

By definition, the error $\mathcal{E}_{t,\nu_1}$ under $\nu_1$ satisfies

$$
\mathcal{E}_{t,\nu_1} \geq \mathbb{P}_1\left(E\right)\mathbb{E}\left[(U_1 - \hat{U}_t)^2 \,|\, E\right] \geq \mathbb{P}_1\left(E\right)e^{2r_1}\epsilon^2,
$$

and the error $\mathcal{E}_{t,\nu_2}$ under $\nu_2$ satisfies

$$
\mathcal{E}_{t,\nu_2} \geq \mathbb{P}_2\left(E^c\right)\mathbb{E}\left[(U_2 - \hat{U}_t)^2 \,|\, E^c\right] \geq \mathbb{P}_2\left(E^c\right)e^{2r_1}\epsilon^2.
$$

Therefore, under the assumption that the algorithm suffers $O(\frac{1}{t})$ error in both environments,

$$
O(\frac{1}{t}) = \mathcal{E}_{t,P_1} + \mathcal{E}_{t,P_2} \geq \mathbb{P}_1\left(E\right)e^{2r_1}\epsilon^2 + \mathbb{P}_2\left(E^c\right)e^{2r_1}\epsilon^2
$$

$$
= e^{2r_1}\epsilon^2\left(\mathbb{P}_1\left(E\right) + \mathbb{P}_2\left(E^c\right)\right) \geq \frac{1}{2}e^{2r_1}\epsilon^2 e^{-2N_t(k)\epsilon^2}.
$$

where the last inequality follows by Pinsker's inequality and Divergence decomposition Lemma [11]. Therefore $N_t(k) = \Omega(\log(t))$. $\qquad\square$

*Proof of Lemma 3.* Define

$$
\Gamma(a) = \Phi(a) - r(a) = \frac{N_t(a)}{N_t(a) + \tau_0}(\hat{r}(a) - r(a)) + \frac{1/2 - \tau_0 r(a)}{\tau_0 + N_t(a)}.
$$

By the fact that the variance of $X_{a,t}$ given $\mathbf{r}$ is 1,

$$
\mathbb{E}\left[\Gamma(a) \,|\, N_t(a), \mathbf{r}\right] = \frac{1/2 - \tau_0 r(a)}{\tau_0 + N_t(a)}.
$$

$$
\mathbb{E}\left[\Gamma(a)^2 \,|\, N_t(a), \mathbf{r}\right] = \frac{\sigma^2 N_t(a)}{(N_t(a) + \tau_0)^2} + O\left(\frac{1}{N_t^2(a)}\right),
$$

Then we have

$$\mathbb{E}\left[\Delta(a)|N_t(a),\mathbf{r}\right] = e^{r(a)} - \mathbb{E}\left[e^{\Phi(a)}|N_t(a),\mathbf{r}\right]$$

$$= e^{r(a)}\left(1 - \mathbb{E}\left[e^{\Gamma(a)}|N_t(a),\mathbf{r}\right]\right) = O\left(\frac{1}{N_t(a)}\right)$$

Similarly,

$$\mathbb{E}\left[\Delta(a)^2 \mid N_t(a),\mathbf{r}\right] = e^{2r(a)}\left(\frac{N_t(j)}{(N_t(j)+\sigma_0)^2} + O\left(\frac{1}{N_t^2(j)}\right)\right).$$

$\square$

*Proof of Lemma 4.* By the choice of $\lambda_s = \frac{|\mathcal{A}|}{\log(1+s)}$, it follows that for all $a$ and $t \geq 4$,

$$\tilde{N}_t(a) = \sum_{s=1}^{t} \pi_s(a) \geq \sum_{s=1}^{t} \frac{1}{\log(1+s)}$$

$$\geq \sum_{s=1}^{t} \frac{1}{\log(1+s)} - \frac{s/(s+1)}{(\log(1+s))^2}$$

$$\geq \int_{1}^{1+t} \frac{1}{\log(1+s)} - \frac{s/(s+1)}{(\log(1+s))^2} ds$$

$$= \frac{1+t}{\log(2+t)} - \frac{1}{\log 2}$$

$$\geq \frac{t}{2\log(2+t)}$$

Conditioned on the event $\widetilde{E}_\epsilon$ where we set $\epsilon = \frac{t}{4\log(2+t)}$, it follows that $N_t(a) \geq \frac{t}{4\log(2+t)}$. Then, for any action $a$ by the definition of sub-gaussian,

$$\mathbb{P}\left(|r(a) - \hat{r}_t(a)| > \sqrt{\frac{8\sigma^2 \log(\frac{2}{\delta})\log(2+t)}{t}}\right)$$

$$\leq \mathbb{P}\left(|r(a) - \hat{r}_t(a)| > \sqrt{\frac{2\sigma^2 \log(\frac{2}{\delta})}{N_t(a)}}\right) \leq \delta.$$

Let $\delta$ satisfy that $\log(2/\delta) = \frac{t}{(\log(2+t))^3}$,

$$\mathbb{P}\left(|r(a) - \hat{r}_t(a)| > \frac{2\sigma}{\log(2+t)}\right) \leq 2\exp\left(-\frac{t}{(\log(2+t))^3}\right)$$

Therefore for $t \geq 2$

$$\mathbb{P}\left(\|\mathbf{r}_t - \hat{\mathbf{r}}_t\|_\infty \geq \frac{2\sigma}{\log(2+t)}\right)$$

$$\leq \mathbb{P}\left(\|\mathbf{r}_t - \hat{\mathbf{r}}_t\|_\infty \geq \frac{2\sigma}{\log(2+t)} \,\bigg|\, \widetilde{E}_\epsilon\right) + \mathbb{P}\left(\widetilde{E}_\epsilon^c\right)$$

$$\leq \sum_k \mathbb{P}\left(|r(a) - \hat{r}_t(a)| > \frac{2\sigma}{\log(2+t)} \,\bigg|\, \widetilde{E}_\epsilon\right) + \mathbb{P}\left(\widetilde{E}_\epsilon^c\right)$$

$$\leq 2|\mathcal{A}|\exp\left(-\frac{t}{(\log(2+t))^3}\right) + 2|\mathcal{A}|\exp\left(-\frac{t}{2(\log(t+2))^2}\right)$$

$$\leq 4|\mathcal{A}|\exp\left(-\frac{t}{(\log(2+t))^3}\right)$$

$\square$

*Proof of Lemma 5.* Note that

$$\left\|\pi^{(1)} - \pi^{(2)}\right\|_\infty \leq \left\|\log \pi^{(1)} - \log \pi^{(2)}\right\|_\infty$$

$$\leq \frac{1}{\tau}\left\|\mathbf{r}^{(1)} - \mathbf{r}^{(2)}\right\|_\infty + \left|\mathcal{F}_\tau(\mathbf{r}^{(1)}) - \mathcal{F}_\tau(\mathbf{r}^{(2)})\right|$$

The proof ends by using the fact $\left|\mathcal{F}_\tau(\mathbf{r}^{(1)}) - \mathcal{F}_\tau(\mathbf{r}^{(2)})\right| \leq \left\|\mathbf{r}^{(1)} - \mathbf{r}^{(2)}\right\|_\infty$, which follows Lemma 8 of [12]. □

# C  Proofs for Tree

This section contains the detailed proof for theorems in the tree setting, in particular, Theorem 4 and Theorem 5.

## C.1  Proof of Theorem 4

*Proof.* We prove this using induction on the depth $D$ of tree. For the base case (D=0), the result directly follows by the fact $\nu$ is sub-gaussian. Now, at some internal node $n(s) \in \mathcal{T}$, assume the result holds for all its children, we prove the result still holds.

For any state $s$, we define $\text{EV}(s) = \exp(V_{\text{sft}}(s)/\tau)$ and $\text{EV}^*(s) = \exp(V_{\text{sft}}^*(s)/\tau)$. Note that

$$\text{EV} - \text{EV}^* \geq \epsilon\text{EV}^* \Leftrightarrow V \geq \tau\log(1+\epsilon) + V^*$$

$$\text{EV}^* - \text{EV} \geq \epsilon\text{EV}^* \Leftrightarrow V \leq \tau\log(1-\epsilon) + V^*$$

Therefore it is equivalent to prove for any node in tree,

$$\mathbb{P}\left(|\text{EV}(s) - \text{EV}^*(s)| \geq \epsilon\text{EV}^*(s)|E_s\right) \leq \tilde{C}\exp\left\{-\frac{\epsilon^2 N(s)}{C\sigma^2}\right\}$$

for some constant $C$ and $\tilde{C}$. Note that by the definition of $U$ we have

$$\text{EV}(s) = \sum_a \exp(Q_{\text{sft}}(s,a)/\tau) = \sum_a \exp\{(r(s,a) + V_{\text{sft}}(s_a))/\tau\}$$

where $s_a$ is the state reached by taking action $a$ at state $s$. Since the reward is deterministic and bounded which only affects the scale, we can then only consider the convergence of $V_{\text{sft}}(s_a)$. Consider a decompose vector $\alpha$ such that $\sum_a \alpha_a\text{EV}^*(s_a) = \epsilon\text{EV}^*(s)$.

$$\mathbb{P}\left(|\text{EV}(s) - \text{EV}^*(s)| \geq \epsilon\text{EV}^*(s) \,|\, E_s\right) \leq \sum_a \mathbb{P}\left(|\text{EV}(s_a) - \text{EV}^*(s_a)| \geq \alpha_a\text{EV}^*(s_a) \,|\, E_s\right)$$

$$\leq \sum_a \tilde{C}_a\exp\left(-\frac{\alpha_a^2 N(s)\pi_{\text{sft}}^*(a|s)}{2C_a\sigma^2}\right),$$

where the last inequality is by the induction hypothesis. Let $\alpha_a^2 \pi_{\text{sft}}^*(a|s) = M$ where $\sqrt{M} = \frac{\epsilon \text{EV}^*(s)}{\sum_a \text{EV}^*(s_a)/\sqrt{\pi_{\text{sft}}^*(a|s)}}$. One can verify that $\sum_a \alpha_a \text{EV}^*(s_a) = \epsilon \text{EV}^*(s)$. Therefore,

$$\mathbb{P}\left(|\text{EV}(s) - \text{EV}^*(s)| \geq \epsilon \text{EV}^*(s)\right) \leq \sum_a \tilde{C}_a \exp\left(-\frac{N(s)}{2C_a\sigma^2}\left(\frac{\epsilon \text{EV}^*(s)}{\sum_a \text{EV}^*(s_a)/\sqrt{\pi_{\text{sft}}^*(a|s)}}\right)^2\right)$$

$$\leq |\mathcal{A}|\tilde{C}\exp\left(-\frac{\epsilon^2 N(s)}{2C\sigma^2}\frac{\text{EV}^*(s)^2}{\left(\sum_a \sqrt{\text{EV}^*(s)\text{EV}^*(s_a)}\right)^2}\right)$$

$$\leq |\mathcal{A}|\tilde{C}\exp\left(-\frac{\epsilon^2 N(s)}{2C\sigma^2}\frac{\text{EV}^*(s)}{\left(\sum_a \sqrt{\text{EV}^*(s_a)}\right)^2}\right)$$

$$\leq |\mathcal{A}|\tilde{C}\exp\left(-\frac{1}{|\mathcal{A}|}\frac{\epsilon^2 N(s)}{2C\sigma^2}\right)$$

$$\leq \tilde{C}_1 \exp\left(-\frac{\epsilon^2 N(s)}{\tilde{C}_2\sigma^2}\right).$$

Picking $\tilde{C} = \max\{\tilde{C}_1, \tilde{C}_2\}$ leads to the conclusion. $\qquad\square$

### C.2 Proof of Theorem 5

*Proof.* Let $a^*$ be the action with largest softmax value and $s$ be the root state. Moreover, let $U(s_a) = \exp\left(Q_{\text{sft}}(s,a)/\tau\right)$ and $U^*(s_a) = \exp\left(Q_{\text{sft}}^*(s,a)/\tau\right)$. The event $E_s$ is defined as in Theorem 4. The probability that MENT selects an sub-optimal arm at $s$ is,

$$\mathbb{P}\left(\exists a \in \mathcal{A}, U(s_a) > U(s_{a^*})\right) \leq \mathbb{P}\left(\exists a \in \mathcal{A}, U(s_a) > U(s_{a^*}) \,|\, E_s\right) + \mathbb{P}\left(E_s^c\right)$$

$$\leq \sum_a \mathbb{P}\left(U(s_a) > U(s_{a^*}) \,|\, E_s\right) + \mathbb{P}\left(E_s^c\right).$$

Since we can upper bound $\mathbb{P}\left(E_s^c\right)$ by Theorem 3, it remains to bound $\mathbb{P}\left(U(s_a) > U(s_{a^*}) \,|\, E_s\right)$.

$$\mathbb{P}\left(U(s_a) > U(s_{a^*}) \,|\, E_s\right)$$

$$= \mathbb{P}\left(U(s_a) - U(s_{a^*}) - U^*(s_a) + U^*(s_{a^*}) > U^*(s_{a^*}) - U^*(s_a) \,|\, E_s\right)$$

$$\leq \mathbb{P}\left(|U(s_{a^*}) - U^*(s_{a^*})| > \alpha U^*(s_{a^*}) \,|\, E_s\right) + \mathbb{P}\left(|U(s_a) - U^*(s_a)| > \beta U^*(s_a) \,|\, E_s\right)$$

$$\leq \tilde{C}_{a^*} \exp\left\{-\frac{N^*(s,a^*)\alpha^2}{2C_{a^*}\sigma^2}\right\} + \tilde{C}_a \exp\left\{-\frac{N^*(s,a)\beta^2}{2C_a\sigma^2}\right\}$$

where $\alpha U^*(s_{a^*}) + \beta U^*(s_a) = U^*(s_{a^*}) - U^*(s_a)$. The last inequality follows by Theorem 4, since $U(s_a) - U^*(s_a) = \exp(r(s,a))\left(\exp\left(V_{\text{sft}}(s')\right) - \exp\left(V_{\text{sft}}^*(s')\right)\right)$, where $s'$ is the state of the child of $n(s)$ taking action $a$. Recall that for any action $a$, $N^*(s,a) = t \cdot \pi_{\text{sft}}^*(a|s)$. We can choose $\alpha$ and $\beta$ similarly as in the proof,

$$\alpha = \frac{(U^*(s_{a^*}) - U^*(s_a))/\sqrt{\pi_{\text{sft}}^*(a^*|s)}}{U^*(s,a)/\sqrt{\pi_{\text{sft}}^*(a|s)} + U^*(s,a^*)/\sqrt{\pi_{\text{sft}}^*(a^*|s)}}$$

$$\beta = \frac{(U^*(s_{a^*}) - U^*(s_a))/\sqrt{\pi_{\text{sft}}^*(a|s)}}{U^*(s,a)/\sqrt{\pi_{\text{sft}}^*(a|s)} + U^*(s,a^*)/\sqrt{\pi_{\text{sft}}^*(a^*|s)}}.$$

Then, there exists some constant $C_a$ and $C_a'$ such that

$$\mathbb{P}\left(U(s_a) > U(s_{a^*}) \,|\, E_s\right) \leq C_a' \exp\left(-\frac{t}{2C_a\sigma^2}\frac{(U^*(s_{a^*}) - U^*(s_a))^2}{U^*(s)(\sqrt{U^*(s,a)} + \sqrt{U^*(s,a^*)})^2}\right).$$

We can omit the terms depending on $U^*$ since they only affect the scale (we can switch to a new constant $C'_a$.) Finally, by Theorem 3,

$$\mathbb{P}\left(\exists a \in \mathcal{A}, U(s_a) > U(s_{a^*})\right) \leq \sum_a \mathbb{P}\left(U(s_a) > U(s_{a^*}) \,\middle|\, E_s\right) + \mathbb{P}\left(E_s^c\right)$$

$$\leq \sum_a C'_a \exp\left\{-\frac{t}{2C_a\sigma^2}\right\} + C't\exp\left\{-\frac{t}{(\log t)^3}\right\}$$

$$\leq Ct \exp\left\{-\frac{t}{(\log t)^3}\right\}$$

for some constant $C$ not depending on $t$. $\qquad\square$