[Reviews · NeurIPS 2019]

Reviewer 1



This paper proposes a new MCTS algorithm, Maximum Entropy for Tree Search (MENTS), which combines the maximum entropy policy optimization framework with MCTS for more efficient online planning in sequential decision problems. The main idea is to replace the Monte Carlo value estimate with the softmax value estimate as in the maximum entropy policy optimization framework, such that the state value can be estimated and back-propagated more efficiently in the search tree. Another main novelty is that it proposes an optimal algorithm, Empirical Exponential Weight (E2W), to be the tree policy to do more exploration. It shows that MENTS can achieve an exponential convergence rate towards finding the optimal action at the root of the tree, which is much faster than the polynomial convergence rate of the UCT method. The experimental results also demonstrate that MENTS performs significantly better than UCT in terms of sample efficiency, in both synthetic problems and Atari games. Overall, I found this to be an interesting paper that addresses the important problem of developing more sample efficient online planning algorithm. The proposed work is well motivated and the idea is very clear. I enjoyed reading the paper. The main weakness I found in the paper is that the experimental results for Atari games are not significant enough. Here are my questions: - In the proposed E2W algorithm, what is the intuition behind the very specific choice of $\lambda_t$ for encouraging exploration? What if the exploration parameter $\epsilon$ is not included? Also, why is $\sum_a N(s, a)$ (but not $N(s, a)$) used for $\lambda_s$ in Equation (7)? - In Figure 3, when $d=5$, MENTS performs slightly worse than UCT at the beginning (for about 20 simulation steps) and then suddenly performs much better than UCT. Any hypothesis about this? It makes me wonder whether the algorithm scales with larger tree depth $d$. - In Table 1, what are the standard errors? Is it just one run for each algorithm? There is no learning curve showing whether each algorithm converges. What about the final performance? It’s hard for me to justify the significance of the results without these details. - In Appendix A (experimental details), there are sentences like The exploration parameters for both algorithms are tuned from {}.’’ What are the exact values of all the hyperparameters used for generating the figures and tables? What hyperparameters is the algorithm sensitive to? Please make it more clear to help researchers replicate the results. To summarize based on the four review criteria: - Originality: To the best of my knowledge, the algorithm presented is original: it builds on previous work (a combination of MCTS and maximum entropy policy optimization), but comes up with a new idea for selecting actions in the tree based on the softmax value estimate. - Quality: The contribution is technically sound. The proposed method is shown to achieve an exponential convergence rate to the optimal solution, which is much faster than the polynomial convergence rate of UCT. It is also evaluated on two test domains with some good results. The experimental results for Atari games are not significant enough though. - Clarity: The paper is clear and well-written. - Significance: I think the paper is likely to be useful to those working on developing more sample efficient online planning algorithms. UPDATE: Thanks for the author's response! It addresses some of my concerns about the significance of the results. But it is still not strong enough to cause me to increase my score as it is already relatively high.

Reviewer 2



This paper presents an extension to MCTS that uses maximum entropy/softmax instead of UCB and hard maximization. A bandit-based (horizon 1) and tree-based version of the algorithm are given along with theoretical results showing improved convergence rate (exponential instead of polynomial). Experiments are also given that show the proposed method (Maximum Entropy Tree Search or MENTS) outperforms UCT on a synthetic tree benchmark and a set of Atari games. MCTS and UCT are widely used, but as pointed out by the paper, the convergence rate is slow and practical performance is not always well understood. Therefore, developing improved, more efficient algorithms is an important problem. Maximum entropy methods have become popular in RL, so the idea of using them with MCTS is promising. The algorithm itself is somewhat straightforward: the softmax value is optimized to choose actions within the tree (instead of UCB) and softmax values are used (and backed up) at the leaves (instead of 'hard' max values). This makes MENTS relatively simple to implement, but potentially quite powerful. The main contribution of the paper is the idea to combine maximum entropy methods and MCTS along with a proof of convergence in the bandit and tree (i.e., sequential) case. The tree case follows from the bandit case and the efficiency improvements are significant (exponential convergence rather than polynomial). The theorems are included in the paper and proofs are included in the supplement. These proofs are a significant contribution to the state-of-the-art in sample-based planning. The experiments show that the method works in practice as well. It is nice that MENTS outperforms UCT on the synthetic tree domain, but more significant that it does so on some of the Atari domains. Unfortunately, there is little discussion about the results and comparison only with 'vanilla' UCT. More discussion and analysis of the online planning results would be helpful since MENTS does not always outperform the other methods. It would have strengthened the paper to also present results for various number of simulations for the Atari games as is done for the sythetic tree domain. Also, as noted in the related work, several variants of UCT have been proposed that may result in better values. It would be helpful to see the results from some subset of these other methods. The writing was generally clear, but a few things could be improved. For example, the axes should be more clearly labeled in Figures 2 and 3 and the 'example' in Figure 1 is not really an example, but a list of the relevant aspects of UCT and MENTS. ==== After rebuttal ==== After the response and discussion, I still believe this is a strong paper. The authors should make the issues from the reviews more clear, but the approach is novel, the theoretical and empirical results are strong and the approach is widely applicable.

Reviewer 3



The authors apply entropy regularization techniques to Monte-Carlo tree search (MCTS). For this purpose, they first investigate bandit problems with the entropy regularization and introduce a new bandit setting called a stochastic softmax bandit problem. The problem is to optimize the value of the expected value plus the entropy regularization term, instead of the (pure) expected reward. Then an optimal action-selection strategy called E2W is proposed for this problem. By applying E2W to the tree search, a new MCTS method is proposed. The paper provides theoretical analysis regarding the convergence rate of the proposed methods. The result indicates that the proposed tree-search could improve the state-of-the-art tree-search based on UCT. However, I have a concern as noted in [a] bellow. The paper also demonstrates the sample efficiency of the proposed method in both synthetic problems and Atari game playing. Concerns: [a] The authors claim that the proposed method has better convergence rate that UCT. However the optimization problem and condition between them will be different. For example, the existing methods usually consider minimization of the regret or the negative expected reward, while the proposed method consider the expected reward plus the entropy as the objective. Furthermore, the proposed method seems to assume that the reward (and state-transition) are deterministic. I would like to recommend a fair and careful discussion (and experiments) with describing difference of the conditions and the objective functions. [b] While the experimental results simply show that the proposed method could have better performance than the state of the art, there is no discussion. Some discussion and intuitive explanation will be useful for better understanding of the proposed approach. In other words, in which case should we use the proposed methods instead of UCT or the other methods? Also note that the proposed algorithm has an additional hyperparameter tau, which could cause the over-fitting to the tasks. [c] The regularization coefficient tau will have big impacts on the performance because it can change the optimal arm and also effect on the convergence rate. Thus, the setting of tau is important to use the proposed methods. However, there is no (empirical) evaluation regarding tau. For example, it should be useful to evaluate the sensitivity of the tau setting. Minor issues: - The definition of H in eq. (2) is missing. - I feel that the result of UCT in the rightest figure in fig. 3 is strange. Why does the decrease of the error stop? === * Update after the rebuttal I would like to thank the authors for submitting a response to the reviews and addressing some of my comments. I have the following comments. - I agree that MENTS is directly compared with UCT in the experiments of Figure 3. However, the optimal action a* in Theorem 5, which is the action with largest softmax-value rather than the conventional action-value, could be different with the optimal action in the conventional RL or UCT. If it is true, the difference will be important and should be mentioned. - I would like to have more meaningful discussion about the decrease of error of UCT in Figure 3 (k=8,d=5). I feel that the rebuttal to this point is just a paraphrase of the experimental result. - The assumption of deterministic transition and reward in Line53 will be unnecessary because it can be misleading.

[Author Response · NeurIPS 2019]

We thank the reviewers for their valuable feedback and suggestions for improving the paper. We are glad the reviewers believe this paper presents a useful contribution on developing more sample efficient online planning algorithms. Our responses to the reviewers' comments are below.

To R1. The specific choice of $\lambda_t$ is made to guarantee the convergence rate in the tree case. It guarantees that at each node of the tree MENTS performs sufficient exploration for all actions so that the softmax value can still be efficiently estimated even under the drift condition (Theorem 4). In equation (7), the decay rate is decided by the total number of simulations of $s$, which is $N(s) = \sum_a N(s, a)$. Table 1 presents the final performance of MENTS and UCT when using 500 simulations to generate a move. All results are averaged over 10 environment restarts. We will present the standard errors and exact values of all hyperparameters in the final version of the paper. In Figure 3 ($k = 8, d = 5$), the decrease of error of UCT stops because it does not propose any reasonable strategy given the simulation budgets.

To R2. Thanks for your suggestions. We did not compare MENTS with other variants of UCT, such as UCT with TD($\lambda$) backups, since it has not been shown that with those variants the convergence property of vanilla UCT still holds. We will add this comparison in the updated paper.

To R3. We agree that entropy regularization introduces a different learning objective compared with existing methods. However, in both theoretical analysis (Theorem 5) and experiments (Figure 3), we directly compare MENTS with UCT in terms of the efficiency of finding the best action at the root. The reason we propose to minimize the MSE of the estimated softmax value in the *stochastic softmax bandit* setting is because it is difficult to directly apply the regret minimization objective in this setting. In the proposed lower bound (Theorem 1), we show that minimizing the MSE is equal to finding the optimal softmax policy. We would also like to point out that MENTS achieves a better convergence rate under weaker assumptions. Note that UCT assumes the value estimation convergence property of the internal nodes under drift condition (see Section 2.4 of [2]), while MENTS only needs the sub-Gaussianness on the leaves. This is because of the benefit of using the entropy regularization in the tree case, that the softmax value of the internal node is guaranteed to satisfy the sub-Gaussianness (Theorem 4). Furthermore, our proposed method and analysis can be directly and completely applied to the setting with stochastic transitions and rewards using standard techniques, such as those in [1]. For example, we can use empirical estimates of rewards and transitions in the search tree. The convergence rate of the empirical estimations follow from Hoeffding's inequality and therefore will not affect our results. In Figure 3 ($k = 8, d = 5$), the decrease in error of UCT stops because it does not propose any reasonable strategy given the simulation budgets.

We thank all reviewers for pointing out the need to strengthen the discussion, particularly in the experimental section. We will add more discussion about the effect of entropy regularization and analysis of the online planning experiment results in the revised version of the paper. We will also conduct more experiments, including sensitivity test of parameter $\tau$ and comparison with UCT variants.

## References

[1] Michael Kearns, Yishay Mansour, and Andrew Y Ng. A sparse sampling algorithm for near-optimal planning in large markov decision processes. *Machine learning*, 49(2-3):193–208, 2002.

[2] Levente Kocsis and Csaba Szepesvári. Bandit based monte-carlo planning. In *European conference on machine learning*, pages 282–293. Springer, 2006.


[Meta-Review · NeurIPS 2019]

This paper presents an appealing idea to combine current max-entropy methods in RL with Monte-Carlo Tree Search. A theoretical result shows improved rate of convergence, while empirical results show improved sample efficiency. The initial reviews were quite positive; I only noted a small number of issues mentioned in the reviews of R1 and R3. In our discussions after reading the author feedback, R3 noted that some of his concerns have not been addressed. R2 replied, saying that these concerns are relatively minor and can be addressed in the final version. With final scores of (8, 7, 6) the paper has quite good chances being accepted NeurIPS.